# Barriers and facilitators to integrating tuberculosis treatment into community pharmacies for people with TB/HIV in Kampala, Uganda: A qualitative study

Jonathan Izudi[1,2,3]*, Adithya Cattamanchi[4,5], Christine Sekaggya-Wiltshire[2,6,7], Noah Kiwanuka[8], Amanda Sammann[9], Rachel King[10]

1 Directorate of Graduate Training, Research and Innovation, Muni University, Arua, Uganda, 2 Research Department, Makerere University Infectious Diseases Institute, Kampala, Uganda, 3 Department of Community Health, Faculty of Medicine, Mbarara University of Science and Technology, Mbarara, Uganda, 4 Division of Pulmonary Diseases and Critical Care Medicine, University of California Irvine, Irvine, California, United States of America, 5 Center for Tuberculosis, Institute for Global Health Sciences, University of California San Francisco, San Francisco, California, United States of America, 6 Mulago National Referral and Specialized Hospital, Kampala, Uganda, 7 School of Medicine, St. Andrews University, Scotland, United Kingdom, 8 Department of Epidemiology and Biostatistics, School of Public Health, Makerere University, Kampala, Uganda, 9 Department of Surgery, University of California San Francisco (UCSF), California, United States of America, 10 Institute for Global Health Sciences, University of California San Francisco (UCSF), California, United States of America

* jonahzd@gmail.com

## Abstract

Community pharmacies (private retail drug shops or pharmacies) have proven successful for delivering anti-retroviral therapy to people living with human immunodeficiency virus (HIV). Leveraging this model for tuberculosis (TB) treatment integration could improve access to both TB and HIV treatment among people with TB/HIV. We identified the barriers and facilitators to integrating TB treatment into community pharmacies for people with TB/HIV in Kampala, Uganda, using a qualitative study across six public health facilities. Participants included people with TB/HIV; healthcare providers (HCPs) from the six facilities and their affiliated community pharmacies; and experts from the Ministry of Health. Data were collected through interviews guided by the Consolidated Framework for Implementation Research (CFIR). We performed framework analysis and mapped the emergent sub-themes to the relevant CFIR domains. We enrolled 47 participants: six TB focal persons, six HIV focal persons, nine pharmacy HCPs, three Ministry of Health staff, and 23 people with TB/HIV. Major facilitators included the convenience of accessing both TB and HIV treatment at a single location; enhanced privacy and reduced stigma; improved accessibility through extended hours, shorter wait times, and proximity; readiness of community pharmacy HCPs to deliver TB treatment; willingness of people with TB/HIV to engage in self-managed care; and training of pharmacy HCPs in TB care. Key barriers included unclear eligibility criteria for enrolling people with TB/HIV, absence

**Data availability statement:** All relevant data are within the paper.

**Funding:** This work was supported by the Fogarty International Center of the National Institutes of Health (K43TW012839, awarded to JI). The content is solely the responsibility of the authors and does not necessarily represent the official views of the National Institutes of Health. The funders had no role in study design, data collection and analysis, decision to publish, or preparation of the manuscript.

**Competing interests:** The authors have declared that no competing interests exist.

of TB counseling services at pharmacies, inadequate infrastructure for TB drug storage, limited confidence among pharmacy HCPs in delivering TB care, and unclear logistics and operational procedures for implementation. Therefore, context-specific interventions that are developed in collaboration with key stakeholders, address barriers like eligibility criteria refinement, enhanced pharmacy HCP training, and financial incentives, and leverage facilitators like policy support and stakeholder readiness, are needed for the successful integration of TB treatment into community pharmacies for people with TB/HIV in Uganda.

## Introduction

In 2023, 10.8 million people were diagnosed with tuberculosis (TB) globally, and 6.1% of those resulted from Human Immunodeficiency Virus (HIV) [1]. TB and HIV are among the leading causes of death globally [2], with TB being the main cause of death in people living with HIV (PLHIV) [3,4]. Conversely, HIV negatively impacts treatment outcomes among people with TB. Systematic reviews show a 70–71% pooled treatment success rate (TSR) among people with TB/HIV [5–7], which is suboptimal compared to the World Health Organization (WHO) desired target of ≥90%. Additionally, people with TB/HIV have a 35–47% lower TSR [7,8] and a 24% higher risk of multidrug-resistant TB [9] compared to those without HIV. In Uganda, the TSR among people with TB/HIV is approximately 70% [10], falling below the WHO's desired target.

HIV significantly contributes to suboptimal TSR among people with TB, primarily due to immune suppression, which compromises the body's ability to fight TB. Additional factors, such as drug interactions, higher pill burden, and opportunistic infections, also play a significant role [11]. Furthermore, longer travel distances that lead to missed medication refill appointments reduce TSR among people with TB/HIV [12,13]. A study in rural eastern Uganda found that people with TB/HIV who travel ≥5 km to a TB clinic have a 17% lower TSR compared to those who travel <5 km to the same clinic [14]. In Kampala, people with TB and/or HIV who travel ≥2 km to a TB clinic have 9–27% higher mortality compared to those who travel <2km to the same clinic [15]. These findings suggest a need for novel and innovative strategies to improve TSR among people with TB/HIV [16].

Community pharmacies, also referred to as private retail drug shops, represent a person-centered differentiated service delivery (DSD) model in Uganda, currently used exclusively to provide anti-retroviral therapy (ART) refills for PLHIV. Studies have shown that community pharmacies have substantial benefits among PLHIV, including enhancing ART access and coverage [17], ensuring convenient ART refills, and reducing stigma [18]. Additionally, through community pharmacies, studies have shown that PLHIV achieved ≥95% ART adherence [19] as well as ≥95 viral load suppression [20]. Healthcare providers (HCPs) have reported decongestion at the health facilities due to reduced patient load [21] and time saved, allowing them to focus on PLHIV who need more care and those with ART-related adherence challenges

[17,21]. Benefits previously documented among PLHIV receiving ART through community pharmacies could similarly extend to those with TB/HIV if TB drug refills are integrated into these pharmacies.

The integration of TB treatment into community pharmacies would enable convenient access to both ART and TB medication refills at the same location and time. However, uncertainties remain regarding contextual factors that may influence the successful implementation of TB treatment integration in this setting. Accordingly, we designed the Community Pharmacy Tuberculosis Treatment (COPHAT) study to explore the barriers and facilitators to integrating TB treatment into community pharmacies among people with TB/HIV in Kampala, Uganda. The evidence will inform the design of context-relevant implementation strategies for integrating TB treatment into community pharmacies for people with TB/HIV in Kampala, Uganda, including testing the intervention in a pilot, type 2 hybrid effectiveness-implementation, individually randomized trial.

## Methods

### Study design and setting

We conducted a theory-informed qualitative study at six primary healthcare facilities in Kampala, the capital city of Uganda. The health facilities provide TB and HIV care per the Uganda national treatment guidelines and operate from 8:00 a.m. to 5:00 p.m., Monday through Friday. Each health facility has separate, dedicated TB and ART clinics, each overseen by a focal person. The focal persons are typically clinical officers, nurses, or medical officers with substantial experience in TB or HIV care. The characteristics and setup of the health facilities have been described in our previous studies [22–24].

### Study population and sampling

Participants included people with TB/HIV who accessed care at the six health facilities; community pharmacy healthcare providers (HCPs) affiliated with each health facility; TB and HIV focal persons from the respective clinics; and officials from the Uganda Ministry of Health (MoH) involved in DSD models. Eligible people with TB/HIV were aged 18 years or older, had been on both TB and HIV treatment for at least two months, and had received ART refills through either a community pharmacy or a health facility. The two-month treatment threshold was set to ensure the participants had sufficient experience with community pharmacy or health facility-based medication refills and to provide meaningful insights. Eligible community pharmacy HCPs, TB and HIV focal persons, and MoH officials (included as DSD Model Experts) had at least six months of experience in TB care or management.

During the study period, 4–6 people with TB/HIV at each TB clinic were consecutively sampled (eligible individuals who presented at the clinic and consented to participate in the study were enrolled).

Focal persons and MoH officials were purposively selected as key informants due to their extensive knowledge and experience in implementing the community pharmacy model. For each health facility, one affiliated community pharmacy serving at least 50 PLHIV was randomly selected. At the selected community pharmacy, 1–2 HCPs were purposively sampled and interviewed. Community pharmacy HCPs interact daily with people with T/HIV, so they were considered to have extensive experience in pharmacy-based drug refills, including anti-retroviral drugs (ARVs).

### Study variables

The study was guided by the Consolidated Framework for Implementation Research (CFIR), an implementation science framework, to identify the implementation determinants of TB treatment integration into community pharmacies. CFIR is also effective for selecting implementation strategies that overcome barriers to intervention implementation and maximize the facilitators [25]. CFIR was also used to guide the development of the data collection tools (S1 File). CFIR consists of five domains that influence implementation either positively or negatively, and we examined all five domains in this study (Table 1).

**Table 1. Summary and description of all CFIR domains and how they relate to the study.**

| CFIR domain | Description and examples |
|---|---|
| Intervention characteristics | This domain referred to the features of the intervention itself—integrating TB treatment into community pharmacies. For example, participants discussed whether flexible medication pick-up times, reduced clinic visits, or convenient pharmacy locations would make the intervention more acceptable or feasible. |
| Characteristics of individuals | This domain covered personal-level factors that may influence the implementation of the intervention. These included clients' stability on ART, adherence to appointments, proximity of residence to pharmacies, presence of treatment support systems, and the knowledge, beliefs, and motivation of pharmacy staff. |
| Inner setting | This domain captured the structural and cultural context within community pharmacies and associated health facilities. It included institutional readiness, communication networks, and the overall implementation climate. For example, how factors such as trained staff, referral mechanisms, and storage capacity for TB medications may influence the integration efforts. |
| Outer setting | This domain encompassed external influences such as policies, patient needs, and community dynamics. For instance, participants described how perceived stigma and discrimination may shape the acceptability of receiving TB treatment from pharmacies. The domain also described the role of the national TB/HIV policies and guidance from regulatory bodies like the Ministry of Health. |
| Process | This domain refers to the activities that may be undertaken to implement the intervention, including planning, engagement, execution, and evaluation. Key elements included coordination between health facilities and pharmacies, integration of TB-related data into health management information systems, sputum follow-up testing, monitoring of treatment outcomes, provision of logistical support, staff training, and supportive supervision. |

## Data collection

Between March 01, 2025, and April 25, 2025, three trained and daily supervised research assistants (one female [NN] and two males [KR and RZ]) conducted in-depth interviews with people with TB/HIV and key informant interviews with the rest of the participants. The research assistants hold bachelor's degrees in health and social sciences and have five or more years of experience in qualitative data collection. The supervision was done by the Principal Investigator through impromptu visits and follow-up at the study sites. In-depth interviews were conducted in *Luganda,* the local language, and lasted approximately 30–60 minutes. Key informant interviews, conducted in English, lasted 30–45 minutes. All interviews were conducted in quiet, convenient locations using a hand-held audio recorder (Sony PCM-A10). Field notes were taken during the interviews to capture contextual details, non-verbal cues, and researcher reflections to complement and enrich the interview data. At the end of each interview, research assistants summarized the major emergent issues, which were used to brief the principal investigator and inform the ongoing direction of the study. During the first week of data collection, they held daily reflection sessions to discuss the interview process, followed by twice-weekly sessions thereafter. These debriefings facilitated peer learning and enabled the principal investigator to provide timely technical support.

## Statistical methods: sample size and data analysis

We determined the sample size using the saturation principle, defined as the point at which additional interviews yield minimal new information [26]. Specifically, after conducting an initial base of 20 interviews, each subsequent interview was assessed for new concepts or themes. Saturation was considered achieved when ≤5% of the information from new interviews was novel, based on a run length of 15, indicating that most themes had already been captured in previous interviews [27]. This approach allowed us to systematically evaluate when sufficient data had been collected.

The interviews were transcribed verbatim by two qualitative researchers (NN and KR) with five or more years of experience in qualitative data transcription. Both transcribers checked the transcripts for accuracy by replaying and reviewing

the respective audio recordings. Inconsistencies were identified and corrected before uploading the transcripts to NVivo version 14.23.3 (Qualitative Data Analysis software) for analysis. To ensure integrity in the transcripts, the Principal Investigator randomly sampled a few transcripts and assessed them for accuracy by replaying the corresponding audio recordings. A framework analysis that employed both deductive and inductive coding approaches, guided by the CFIR domains, was employed. Here, two independent data analysts (JI and RK) conducted the coding and analysis of the transcripts. JI, a male public health specialist with five years of qualitative research experience, and RK, a female socio-behavioral research scientist with over 10 years of research experience, each holds a doctorate in public health. JI and RK have expertise in qualitative data collection, analysis, and interpretation.

JI and RK independently reviewed 5–7 transcripts to familiarize themselves with the data, followed by independent coding of the transcripts by flagging texts considered important based on CFIR domains. Emerging patterns were identified from these texts, guided by CFIR constructs, and a preliminary codebook was jointly developed by consensus. The preliminary codebook was then used to code the remaining transcripts, with the flexibility to incorporate new codes as they emerged. Similar codes were grouped into categories, which were then organized into sub-themes aligned with the CFIR domains. Two senior researchers (AC and CW) reviewed and validated the final codebook to ensure analytical rigor. The findings were presented using illustrative quotes and anonymized participant identifiers. Reporting of the findings adhered to the Consolidated Criteria for Reporting Qualitative Research (COREQ) guidelines (S2 File) [28].

## Quality control and scientific rigor

In March 2025, we conducted a three-day training for the research assistants covering the study protocol, responsible conduct of research, and interviewing and probing techniques. The training included structured sessions and six rounds of role plays, which were observed and assessed by the study investigators to ensure proficiency.

We pre-tested the interview guides at the Infectious Diseases Institute TB clinic with participants who met similar characteristics to our study population. Feedback was obtained and used to revise question phrasing, improve flow, and ensure cultural appropriateness. To enhance scientific rigor, we employed triangulation of: 1) data sources, by interviewing participants from different sites; 2) investigators, by using multiple coders during analysis; and 3) methods, by conducting both in-depth and key informant interviews. We also maintained a detailed audit trail of study design decisions, interview notes, coding iterations, and analytic memos to ensure dependability and confirmability. Thick descriptions were used to enable transferability by providing rich context around participants' experiences. We did not conduct member checking due to time constraints, but mitigated this by peer debriefing among the study team and reflexive journaling to enhance credibility and reduce bias.

## Ethical issues

We obtained administrative clearance from the Directorate of Public and Environmental Health, Kampala Capital City Authority (Ref: DPHE/KCCA/1301/01). Ethical approval was granted by the Makerere University Infectious Diseases Institute Research Ethics Committee (Ref: IDI-REC-2024–98) and the Uganda National Council for Science and Technology (Ref: HS4397ES). All participants provided informed consent—either written or by thumbprint—after receiving detailed study information, including the study rationale, purpose, and potential risks and benefits. Additional study information focused on measures to ensure the privacy and confidentiality of information, the right to withdraw at any time, compensation for time spent in the study, and the acquisition of the required ethical approvals.

## Inclusivity in global research

Information regarding ethical, cultural, and scientific considerations specific to inclusivity in global research is included in the Supporting Information (S3 File).

## Results

### Description of participant's characteristics

Table 2 presents the characteristics of the participants included in the study displayed by the type of participants. We enrolled 47 individuals: six TB focal persons, six HIV focal persons, nine community pharmacy HCPs, three MoH experts, and 23 people with TB/HIV. The mean age varied by professional cadre, from 30.7 (standard deviation [SD]=7.0) years among community pharmacy HCPs to 47.7 (SD=3.8) years among MoH experts. Most of the HCPs were younger than 35 years, while people with TB/HIV were more often 35 years or older. Sex distribution was nearly balanced across the groups, though people with TB/HIV were slightly more often male. Educational attainment and employment differed, with all HCPs formally employed and highly educated, while people with TB/HIV reported lower levels of education and were mainly self-employed or unemployed.

**Table 2. Distribution of participants' characteristics by the type of participant.**

| Variables and levels | All (n=47) | HIV Focal persons (n=6) | Community Pharmacy HCPs (n=9) | MoH DSD Model Experts (n=3) | Person with TB/HIV (n=23) | TB Focal persons (n=6) |
|---|---|---|---|---|---|---|
| **Study sites** | | | | | | |
| Kawaala Health Center IV | 8 | 1 | 2 | 0 | 4 | 1 |
| Kisenyi Health Center III | 8 | 1 | 2 | 0 | 4 | 1 |
| Kisugu Health Center III | 6 | 1 | 0 | 0 | 4 | 1 |
| Kiswa Health Center III | 7 | 1 | 2 | 0 | 3 | 1 |
| Kitebi Health Center III | 8 | 1 | 2 | 0 | 4 | 1 |
| Komamboga Health Center III | 7 | 1 | 1 | 0 | 4 | 1 |
| MoH | 3 | 0 | 0 | 3 | 0 | 0 |
| **Age group (years)** | | | | | | |
| Less than 35 | 25 | 5 | 8 | 0 | 9 | 3 |
| 35 and over | 22 | 1 | 1 | 3 | 14 | 3 |
| Mean (standard deviation) | 35.6 (8.5) | 33.2 (4.5) | 30.7 (7.0) | 47.7 (3.8) | 35.6 (8.5) | 39.2 (9.9) |
| **Sex** | | | | | | |
| Female | 22 | 3 | 4 | 2 | 10 | 3 |
| Male | 25 | 3 | 5 | 1 | 13 | 3 |
| **Level of education** | | | | | | |
| None | 4 | 0 | 0 | 0 | 4 | 0 |
| Primary | 10 | 0 | 0 | 0 | 10 | 0 |
| Secondary and over | 33 | 6 | 9 | 3 | 9 | 6 |
| **Qualification** | | | | | | |
| None | 8 | 0 | 0 | 0 | 8 | 0 |
| Certificate | 11 | 0 | 0 | 0 | 11 | 0 |
| Diploma | 7 | 0 | 0 | 0 | 2 | 5 |
| Bachelor's degree | 18 | 6 | 9 | 0 | 2 | 1 |
| Master's degree | 3 | 0 | 0 | 3 | 0 | 0 |
| **Employment** | | | | | | |
| None | 3 | 0 | 0 | 0 | 3 | 0 |
| Self | 17 | 0 | 0 | 0 | 17 | 0 |
| Formal | 27 | 6 | 9 | 3 | 3 | 6 |

**Facilitators and barriers to integrating TB treatment into community pharmacies for people with TB/HIV in Kampala, Uganda**

The facilitators and barriers to integrating TB treatment into community pharmacies for people with TB/HIV, based on the CFIR domains, are summarized in Table 3. Many of the facilitators were in the CFIR domain and focused on the characteristics of individuals. Similarly, the majority of the barriers were in the outer setting of the CFIR domain.

**1. Intervention characteristics**

   **Facilitators.**

   **Convenience of accessing both TB and HIV treatment (dual therapy):**   All participants (people with TB and HCPs) stated that integrating TB treatment into community pharmacies would make TB treatment more convenient and accessible for people with TB/HIV in a single place and reduce the need for multiple visits to the health facility or TB and HIV clinics.

   *"Clients (PLHIV) come to pick ARVs [anti-retroviral drugs] and would want to pick TB drugs from here [community pharmacy] too instead of going to the hospital [health facility]." (Community pharmacy HCP, Female).*

   *"If I can pick both TB and HIV medicines from the pharmacy, that will be good because it is near my home." (Person with TB/HIV, Male).*

**Table 3. Facilitators and barriers to integrating TB treatment into community pharmacies for people with TB/HIV in Kampala, Uganda.**

| CFIR domains | Facilitators | Barriers |
|---|---|---|
| 1. Intervention characteristics | • **Convenience of accessing both TB and HIV treatment (dual therapy).**<br>• Use of adherence and viral load data as eligibility criteria for community pharmacy-based TB treatment | • Unclear eligibility criteria for identifying and enrolling people with TB/HIV into community pharmacies. |
| 2. Outer setting | • Existing national DSD model policy supports the integration of TB services into community pharmacies<br>• **Enhanced anonymity at pharmacies reduces stigma, improves comfort, and ensures privacy.** | • Inadequate patient awareness about the treatment of TB in community pharmacies.<br>• Lack of financial incentives for community pharmacy HCPs to offer TB services<br>• Resistance of people with TB/HIV and HCPs to community pharmacy-based TB treatment. |
| 3. Inner setting | • **Convenience: extended hours, reduced wait times, and proximity make community pharmacies more accessible to people with TB/HIV.** | • **Lack of TB counseling services at community pharmacies.**<br>• Inadequate infrastructure for TB drug storage. |
| 4. Characteristics of individuals | • **Optimism about TB integration and belief in its feasibility and impact.**<br>• **Readiness of community pharmacy HCPs to deliver TB treatment.**<br>• **Willingness of people with TB/HIV to engage in integrated and self-managed TB care.** | • Perceived mistreatment and disrespectful care by community pharmacy HCPs.<br>• **Limited confidence in TB care management among community pharmacy HCPs.** |
| 5. Process | • **Training of community pharmacy HCPs in TB treatment.**<br>• Early involvement of key stakeholders in the integration process. | • **Unclear logistics and operational procedures for implementing TB services within community pharmacies** |

**Note:** Bolded texts are the major facilitators and barriers in the study.

   

Community pharmacy HCPs unanimously emphasized that integrating TB treatment aligns with their existing roles, such as dispensing medications for other chronic conditions, including HIV.

*"We can do it [TB treatment] the same way we do for ARVs [Anti-Retroviral Drugs]. We have been dispensing [TB drugs] to them for two months" (TB focal person, Female).*

**Use of adherence and viral load data as eligibility criteria for community pharmacy-based TB treatment:** Participants emphasized using high ART adherence and suppressed viral load to identify eligible people with TB/HIV for treatment at community pharmacies. They noted that such eligibility criteria would ensure only stable individuals transition to community pharmacies, enhancing integration effectiveness and reducing risks linked to complex clinical management outside health facilities.

*"They [community pharmacies] should not consider someone with a high viral load [detectable or unsuppressed viral load]......The clinicians know how to tell who is adhering well and who is not adhering well to consider for the integration." (Person with TB/HIV, Female).*

**Barriers.**
**Unclear eligibility criteria for identifying and enrolling people with TB/HIV into community pharmacies:** Certain participants (focal persons and people with TB/HIV) indicated undefined eligibility criteria as a barrier to the seamless integration of TB treatment into community pharmacies. The participants continued to state that community pharmacies may not suit all people with TB/HIV, and that lacking clear eligibility criteria could hinder integration.

*"There should be recruitment (eligibility) criteria for community pharmacies. You do not just push in everybody (people with TB/HIV) to the pharmacy." (TB focal person, Male).*

## 2. Outer setting

**Facilitators.**
**Existing national DSD model policy supports the integration of TB services into community pharmacies:** The existence of a national DSD model policy supporting the integration of TB services into community pharmacies emerged as a facilitator. The policy was built on previous successes of integrating ART into pharmacies, enabling the MoH's National TB Control Program to develop and launch an integration plan. The plan aims to support the delivery of TB medication refills, screening, and prevention services at the community pharmacy level.

*"It's now the strategic direction [under the DSD model] of the Ministry of Health to integrate services. We wanted to go beyond just dispensing medicine but also do a bit of screening… TB preventive therapy… follow-up… nutrition assessment." (DSD Model Expert, Male).*

The integration of TB services may further be strengthened by the availability of training packages, which are perceived to facilitate a layered, stepwise rollout—from the national to regional, district, health facility, and community pharmacy levels. DSD model experts at the MoH indicated that training materials have already been developed to facilitate the integration of TB treatment as well as non-communicable diseases into community pharmacies.

*"We already developed a package, an integration package. We already developed some integration training materials." (DSD Model Expert, Female).*

*"We finished the national TOT [Training of Trainers] on integration and are now doing regional TOT on integration. We then go to the district and health facilities."* (DSD Model Expert, Female).

**Enhanced anonymity at pharmacies reduces stigma, improves comfort, and ensures privacy:** All participants indicated that integrating TB treatment into community pharmacies may offer people with TB/HIV a more anonymous treatment environment compared to public health facilities. They stated that the approach may avoid the stigma and discrimination often associated with TB and HIV and ensure people with TB/HIV feel less exposed to social judgment.

*"There are people who are wide-mouthed... somebody sees you and goes talking... But here you will move to the [community] pharmacy and pick your drugs and return home without anybody getting to know."* (Person with TB/HIV, Female).

**Barriers.**
**Inadequate patient awareness about the treatment of TB in community pharmacies:** Insufficient understanding of the capacity of community pharmacies to offer TB treatment was perceived as a barrier. Participants noted that if people with TB/HIV are not aware that community pharmacies can provide TB medication refills, they may continue to return to the health facilities for these services.

They highlighted the need for more targeted communication and education for people with TB/HIV to improve their trust and engagement with pharmacy-based care.

*"If the patients [people with TB/HIV] are not sensitized well at the health facility… they might end up going back [to the health facility] … or just remain in space [go nowhere]."* (TB focal person, Female).

**Lack of financial incentives for community pharmacy providers to offer TB services:** It emerged that a lack of financial incentives for the Community Pharmacy HCPs involved in TB medication refills would hinder the integration of TB services into pharmacies.

Some participants noted that, in the absence of incentives, pharmacies may deprioritize TB treatment due to the substantial time and resource demands associated with providing care for people with TB/HIV.

*"Like for ART integration into community pharmacies, the organization gives us 2,000 Uganda shillings (about $0.55) for every patient [PLHIV dispensed ARVs] …... I think the same can be done with TB treatment"* (Community pharmacy HCP, Male).

**Resistance of people with TB/HIV and HCPs to community pharmacy-based TB treatment:** Findings revealed that some people with TB/HIV may prefer to visit public health facilities for their TB treatment. They expressed doubts about the competence and reliability of community pharmacies in providing TB treatment. This barrier was reinforced by some community pharmacy HCPs, who noted that many people with TB/HIV were unfamiliar with the role of community pharmacies in TB care.

*"The ones [HCPs] here [at the public health facilities] have more experience than those [HCPs] at the community pharmacy."* (Person with TB/HIV, Female).

*"Some of them [people with TB] may be new in this system [community pharmacy-based care] … they may not know the benefits of receiving from the community pharmacy. "Of course, there are those who will accept and those who will refuse."* [Community pharmacy HCP, Female).

Some participants believed that pharmacies are unsuitable for TB treatment because they primarily stock over-the-counter medications like painkillers and antimalarials. They perceived that pharmacies lacked the infrastructure necessary to refill TB drugs.

*"Pharmacies sell Panadol [paracetamol tablets—painkillers] and [anti] malarial drugs. I am not sure they are ready for TB treatment." (Person with TB/HIV, Female).*

### 3. Inner setting

**Facilitators.**

**Convenience: Extended hours, reduced wait times, and proximity make community pharmacies more accessible to people with TB/HIV:** Participants noted that the extended working hours, reduced wait times, and geographical proximity of community pharmacies to people with TB/HIV were facilitators for integrating TB treatment. They mentioned that integrating TB treatment into community pharmacies would significantly improve access to both ART and TB medication refills for people with TB/HIV.

*"Most community pharmacies open at 8.30 am [morning hours], some work for 24 hours, unlike here at the health facility where they [HCPs] work between 9.00 am and 3.00 pm." (TB focal person, Male).*

Some of the participants cited long wait times at health facilities due to high patient load and long queues as reasons why community pharmacies are better suited for integrating TB treatment.

*"You might find that in the health facility, there are many people [high patient load], you have to line up [long queues and then wait time] …but in the community [pharmacy], once you go there, you pick your [TB and HIV] drugs and leave" (Person with TB/HIV, Female).*

Additionally, it emerged that community pharmacies were widely distributed, and their proximity to people with TB/HIV was identified as a key facilitator for integrating TB treatment. This aligns with the preferences of people with TB/HIV by addressing access barriers often encountered at public health facilities.

*"There is a [community] pharmacy almost in every trading center. People do not have to travel far, like they do to reach a health facility. (Person with TB/HIV, Male).*

**Barriers.**

**Lack of TB counseling services at community pharmacies:** People with TB/HIV expressed concerns about the lack of counseling services at community pharmacies. While community pharmacies refill ART to PLHIV, it emerged that if an ART refill appointment was missed for a week, individuals were required to seek counseling at the health facility before continuing with the refills.

People with TB/HIV perceived this additional step as burdensome and a potential barrier to integrating TB treatment into community pharmacies.

*"If a person [with HIV] takes more than one week without coming [to the community pharmacy]… then that person is taken back to the other side of the health facility for [treatment adherence and psychosocial] counseling." (Person with TB/HIV, Female).*

Furthermore, participants expressed concerns about the capacity of community pharmacies to provide counseling services to people with TB/HIV in need. This was cited as a barrier to integrating TB treatment into pharmacies.

*"Most of us [community pharmacy HCPs] are not trained to handle TB patients, especially on how to counsel them or manage [medication] side effects." (Community pharmacy HCP, Male).*

**Inadequate infrastructure for TB drug storage:** Concerns were raised about the availability of space to safely store TB medications in pharmacies. Some people with TB/HIV expressed doubts regarding the ability of pharmacies to store their medications appropriately. In contrast, community pharmacy HCPs acknowledged the storage limitations and indicated that it could hinder the integration of TB treatment.

*"I wonder if [community] pharmacies have the right storage for TB drugs. Where will they [community pharmacies] store the TB drugs? Do they even have space?" (Person with TB/HIV, Male).*

*"The [community] pharmacy is very small and does not have the space to store the TB medication safely." (HIV focal person, Female).*

## 4. Characteristics of individuals

### Facilitators.

**Optimism about TB integration and belief in its feasibility and impact:** Participants expressed optimism about integrating TB treatment into community pharmacies, suggesting that the approach could be feasible and has the potential to significantly improve the health outcomes of people with TB/HIV.

*"It [integration of TB treatment into community pharmacies] will work if the medicines [anti-TB drugs] are there, health workers are available, and treatment is available." (Person with TB/HIV, Male).*

*"I am excited to help integrate TB treatment into our [community pharmacy] services because I believe it will improve patient outcomes." (Community pharmacy HCP, Female).*

**Readiness of community pharmacy HCPs to deliver TB treatment:** Several community pharmacy HCPs expressed readiness to support the integration of TB treatment into pharmacies. They felt professionally obligated to contribute to TB care and reported confidence in their ability to provide TB treatment alongside ART for people with TB/HIV.

*"If trained, I would be confident and ready to dispense TB drugs and explain [to people with TB/HIV] how to take them." (Community pharmacy HCP, Male).*

*"I think whoever can dispense these other drugs can also dispense TB drugs, provided that person is well-oriented." (Community pharmacy HCP, Male).*

**Willingness of people with TB/HIV to engage in integrated and self-managed TB care:** People with TB/HIV expressed willingness to actively engage in integrated and self-managed TB treatment through pharmacies. This readiness was identified as a key facilitator for the successful integration and sustained delivery of TB treatment through community pharmacies.

*"As long as the drugs [TB and HIV drugs] are the same, I do not mind where I get them from." (Person with TB/HIV, Female).*

### Barriers.

**Perceived mistreatment and disrespectful care at community pharmacies:** Some people with TB/HIV reported previous experiences of mistreatment and disrespectful behavior in community pharmacies. They indicated

that such experiences undermine the trust of people with TB/HIV and decrease their willingness to participate in integrated care.

*"…you meet some [community pharmacy] health workers who would insult us [people with TB/HIV], disrespect and discriminate against us, like they say, extend very far, I do not want to contract your TB. At least I should go to the hospital [health facility] where they [HCPs] handle me with care [respectful and compassionate care]…" (Person with TB/HIV, Male).*

**Limited confidence in TB care management among community pharmacy HCPs:** Some community pharmacy HCPs expressed low confidence in their ability to safely and effectively manage TB treatment, viewing it as more complex than HIV management.

*"ART [Anti-retroviral Therapy] is easier [to dispense]. TB has so many complications and I am not confident about treating it." (Community pharmacy HCP, Male).*

### 5. Process

#### Facilitators.

**Training of community pharmacy HCPs in TB treatment:** The training of community pharmacy HCPs on TB treatment emerged as a crucial facilitator for the successful integration of TB treatment into pharmacies. All participants (HCPs and people with TB/HIV) emphasized the need to train community pharmacy HCPs to enhance their confidence and competence in managing complex TB treatment regimens.

*"There is a need for you to first train those people [community pharmacy HCPs] on how to implement [dispense TB drugs]." (HIV focal person, Male).*

*"After that process of talking to us [community pharmacy HCPs], the next step should be the refresher training." (Community pharmacy HCP, Female).*

*"Capacity building and patient literacy are key, tell them [people with TB/HIV] what they can and what they cannot get from the [community] pharmacies." (DSD Model Expert, Female).*

**Early involvement of key stakeholders in the integration process:** Participants emphasized the importance of early involvement of key stakeholders in the process of integrating TB treatment. In particular, they cited the involvement of community pharmacy personnel (HCPs and proprietors), people with TB/HIV, and HCPs at the ministry and health facilities. This inclusive approach was considered crucial for fostering ownership and shaping the integration strategy to effectively address the needs and preferences of people with TB/HIV.

*"They should involve us [people with TB/HIV] before starting it [TB treatment integration]. We can help explain to others [people with TB/HIV] about it." (Person with TB/HIV, Male).*

*"We need to ensure that they [pharmacy HCPs and proprietors] are involved right from the beginning. We talk to them, sensitize them." (HIV focal person, Male).*

The existing partnership between community pharmacies and health facilities was highlighted as a central pillar for strengthening stakeholder engagement. In particular, the established collaboration around ART integration was viewed as important for fostering mutual respect and shared responsibility in integrating TB treatment.

*"We already work with the health facility staff to support HIV care [ART refills to PLHIV]. They can support us for TB [treatment integration] too."* (Community pharmacy HCP, Female).

**Barriers.**

**Unclear logistics and operational procedures for implementing TB services within community pharmacies:** Participants identified the absence of clear mechanisms for drug supply and accountability, reporting, and supervision as barriers to integrating TB treatment into community pharmacies. They noted that these gaps could create confusion and operational challenges, potentially undermining the feasibility and sustainability of the integration process.

*"We need a clear implementation plan. Who supplies the drugs [anti-TB drugs]? Who trains us [the community pharmacy HCPs]? Who supervises us?"* (Community pharmacy HCP, Male).

*"How do we ensure that as these pharmacies are given medicine to dispense, they will account for it properly? "That is why we need an electronic system so that the health facilities can gain confidence and get to know that the pharmacy has such and such number of clients [People with TB/HIV]."* (DSD Model Expert, Male).

## Discussion

We explored the facilitators and barriers to integrating TB treatment into community pharmacies for people with TB/HIV in Kampala, Uganda, using a theory-informed implementation science framework—the CFIR. We found that the facilitators and barriers are primarily rooted in the community pharmacy setting, followed by those at the health worker level, and to a lesser extent, among people with TB/HIV.

The study found that integrating TB treatment into community pharmacies is facilitated by the convenience of accessing both TB and HIV treatments, extended operating hours, and reduced wait times. These findings align with previous research demonstrating that community pharmacies (private retail drug outlets) provide convenient access to healthcare, with most operating at least 12 hours per day and throughout the week [29]. The extended hours make pharmacies an important venue for delivering TB services. Additionally, another study reported that the widespread geographic distribution of community pharmacies facilitates timely access to HCPs, supporting treatment adherence and engagement in care [30].

Enhanced anonymity provided by community pharmacies emerged as a key facilitator of TB treatment integration, as it helps reduce stigma, improve comfort, and promote privacy for people with TB/HIV. This is not surprising, as people with TB/HIV often face stigma [31–33], including the risk of inadvertent status disclosure. Community pharmacies can, therefore, act as a shield against these barriers, which can otherwise delay timely access and continuity of TB/HIV treatment. Enhanced anonymity in community pharmacies reduces the fear of stigma and unwanted disclosure, making people with TB/HIV more comfortable to seek care and continue treatment. This sense of privacy fosters psychological safety, encourages timely treatment initiation, and normalizes TB/HIV care alongside other healthcare needs. Together, these mechanisms help strengthen adherence and continuity of care, ultimately improving treatment outcomes. This finding is supported by several studies reporting that community pharmacies may help prevent inadvertent HIV status disclosure [34] and mitigate both self-stigma and community stigma [19,21,35–37].

The willingness and readiness of community pharmacy HCPs to deliver pharmacy-integrated TB treatment to people with TB/HIV emerged as an important facilitator. Additional facilitators included pharmacy HCPs perceiving the integration as an extension of their professional responsibility to support public health initiatives and improve patient outcomes. The finding is consistent with the results of a Malaysian study, which reported that community pharmacists were willing to supervise TB treatment, suggesting a shared sense of professional commitment across different settings [38].

Another facilitator for TB treatment integration into community pharmacy was the early involvement of key stakeholders, namely HCPs at health facilities and community pharmacies, as well as Ministry of Health officials. This process may be reinforced by leveraging existing partnerships and fostering strong collaborations to enhance TB service delivery.

Effective communication and collaboration between community pharmacies, social workers, and clinicians have been seen as associated with improved identification and management of non-compliance with treatment and adverse drug reactions [30]. These findings align with previous research in Tanzania, which identified stakeholder engagement as an important strategy for involving community pharmacies in TB care [39]. Furthermore, our findings align with another study that supported the role of community pharmacies in strengthening national TB control programs, including improving TB management and treatment outcomes [38].

The training of community pharmacy HCPs in TB treatment emerged as a key facilitator for integrating TB treatment into community pharmacies. In particular, training on medication dispensing and psychosocial and treatment adherence counseling was emphasized to enhance the HCPs' knowledge, confidence, and capacity to deliver TB treatment. The findings align with previous research in high TB-burden countries, which identified targeted training as a critical step in strengthening the role of pharmacy HCPs in TB care delivery by enhancing knowledge and confidence [39]. Aligned with our findings, a previous cross-sectional study in Peru showed that community pharmacy healthcare providers are interested in learning more about TB and expanding their involvement in delivering TB care services within their communities [40].

We found several barriers to the integration of TB treatment into community pharmacies, including the lack of clear eligibility criteria, the absence of financial incentives for community pharmacy HCPs, limited awareness among people with TB/HIV, and resistance from both people with TB/HIV and health facility HCPs. The lack of financial incentives and resistance from both people with TB/HIV and health facility HCPs aligns with previous research, which underscores the unique position of community pharmacies as private business entities [39]. Community pharmacies may resist engaging in TB care due to concerns such as increased workload, regulatory requirements, or the stigma associated with TB management [39].

The study showed additional barriers such as inadequate training and low confidence among pharmacy HCPs, limiting their willingness and capacity of community pharmacies to participate in integrating TB treatment. A previous study indicated that the involvement of community pharmacies in TB care can be improved by providing targeted training and offering appropriate incentives [39]. Well-trained pharmacists can contribute to TB care by correctly dispensing TB medications, providing counseling, monitoring treatment adherence, and supporting treatment follow-up in collaboration with the healthcare team [41].

The study findings indicated that negative perceptions of community pharmacy HCPs towards people with TB/HIV, unclear operational procedures, and the lack of TB counseling services are key barriers to integrating TB treatment into community pharmacies.

We did not find published data to support these findings, given that few studies have focused on this topic. However, the finding suggests that addressing these barriers may require a multi-faceted approach, including engaging key stakeholders in policy and program design and building the capacity of pharmacy HCPs in TB management [39]. Also, strengthening the national TB services integration policy frameworks, improving communication between health facilities and community pharmacies, and ensuring logistical support, including drug supply chain systems, may facilitate the successful integration of TB treatment into community pharmacies.

In summary, with a national DSD model policy backing, several perceived barriers to integrating TB services into community pharmacies, such as unclear eligibility criteria, limited patient awareness, resistance among people with TB/HIV and HCPs, and low confidence among pharmacy HCPs in TB management, are likely to be mitigated. The remaining barriers will likely include logistical and operational gaps, particularly data capture and reporting, concerns around incentivizing pharmacy HCPs, and perceived mistreatment of people with TB/HIV at some pharmacies.

## Study strengths and limitations

The study's strengths include the use of CFIR to underscore both the development of the interview guides and the analytical approach. Our study population included all key stakeholders involved in the community pharmacy model, comprising people with TB/HIV, HCPs from community pharmacies and health facilities, and the MoH. Data collection continued until thematic saturation was achieved, ensuring comprehensive coverage of perspectives. We employed rigorous methodological and analytical procedures, and this enhanced the credibility and trustworthiness of the findings.

Study limitations include potential selection bias, which was addressed by purposively including all key stakeholder groups, including people with TB/HIV and HCPs from community pharmacies and health facilities, and the MoH. Recall and interviewer biases may have influenced the findings, but these were minimized through skilled interviewers who received careful interviewer training and the use of standardized interview guides. While qualitative data interpretation can be subjective, the use of CFIR to guide analysis and results presentation, and the rigorous coding procedures, enhanced the reliability of the findings. Finally, the lack of quantitative data may limit triangulation, but the depth of qualitative insights provided a rich understanding of facilitators and barriers to integration.

## Conclusion and recommendation

We identified several facilitators and barriers to integrating TB treatment into community pharmacies for people with TB/HIV. Therefore, context-specific interventions addressing barriers such as eligibility criteria refinement, enhanced pharmacy HCP training, and financial incentives, and leverage facilitators like policy support and stakeholder readiness, will be refined using a using a Human-Centered Design approach in collaboration with key stakeholders for the successful integration of TB treatment into community pharmacies for people with TB/HIV in Uganda and similar settings. These strategies will subsequently be piloted in a Type 2 hybrid effectiveness-implementation trial to evaluate their feasibility, fidelity, acceptability, and preliminary effectiveness.

## Supporting information

**S1 File. Data collection instruments.**
(PDF)

**S2 File. Consolidated Reporting of Qualitative Studies (COREQ) guideline.**
(PDF)

**S3 File. Inclusivity in global health research.**
(PDF)

## Acknowledgments

We thank our Research Assistants for their dedication and effort in data collection. We are grateful to the Kampala Capital City Authority (KCCA) for providing administrative clearance to conduct the study. We acknowledge the Ministry of Health's National Tuberculosis and Leprosy Programme (NTLP) for its guidance and support. We also thank the Infectious Diseases Institute Research Ethics Committee (IDI-REC) and the Uganda National Council for Science and Technology (UNCST) for granting ethical approval. Special thanks go to the Heads of Health Facilities at the study sites for granting site-level clearance and providing valuable support throughout the study. Finally, we are deeply grateful to the study participants for their time, trust, and willingness to share their experiences.

## Author contributions

**Conceptualization:** Jonathan Izudi, Rachel King.

**Data curation:** Jonathan Izudi.

**Formal analysis:** Jonathan Izudi.

**Funding acquisition:** Jonathan Izudi.

**Investigation:** Jonathan Izudi.

**Methodology:** Jonathan Izudi, Adithya Cattamanchi, Christine Sekaggya-Wiltshire, Noah Kiwanuka, Amanda Sammann, Rachel King.

**Supervision:** Adithya Cattamanchi, Christine Sekaggya-Wiltshire, Noah Kiwanuka, Amanda Sammann, Rachel King.

**Validation:** Jonathan Izudi, Adithya Cattamanchi, Christine Sekaggya-Wiltshire, Noah Kiwanuka, Amanda Sammann, Rachel King.

**Visualization:** Jonathan Izudi, Adithya Cattamanchi, Christine Sekaggya-Wiltshire, Noah Kiwanuka, Amanda Sammann, Rachel King.

**Writing – original draft:** Jonathan Izudi, Adithya Cattamanchi, Christine Sekaggya-Wiltshire, Noah Kiwanuka, Amanda Sammann, Rachel King.

**Writing – review & editing:** Adithya Cattamanchi, Christine Sekaggya-Wiltshire, Noah Kiwanuka, Amanda Sammann, Rachel King.

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
