## [Decision Letter · Decision Letter 0]

30 Sep 2025

PGPH-D-25-01867

Barriers and facilitators to integrating tuberculosis treatment into community pharmacies for people with TB/HIV in Kampala, Uganda: a qualitative study

Dear Dr Izudi,

Thank you for submitting your manuscript to PLOS Global Public Health. After careful consideration, we feel that it has merit but does not fully meet PLOS Global Public Health’s publication criteria as it currently stands. Therefore, we invite you to submit a revised version of the manuscript that addresses the points raised during the review process.

We look forward to receiving your revised manuscript.

Kind regards,

Henry Zakumumpa, PhD

Academic Editor

Journal Requirements:

Additional Editor Comments (if provided):

We are pleased to share comments from our two reviewers for your attention. As observed by one of the reviewers there is need for a more detailed description of the qualitative data analysis approach. Also, please strengthen the discussion section to ensure that there is adequate reflection on the implications of your important study on policy and practise.

Please include a point by point response to each of the comments raised by the reviewers so we can move swiftly to a decision.

Reviewers' comments:

Reviewer's Responses to Questions

**Comments to the Author**

1. Does this manuscript meet PLOS Global Public Health’s publication criteria?

Reviewer #1: Yes

Reviewer #2: Yes

2. Has the statistical analysis been performed appropriately and rigorously?

Reviewer #1: Yes

Reviewer #2: N/A

3. Have the authors made all data underlying the findings in their manuscript fully available (please refer to the Data Availability Statement at the start of the manuscript PDF file)?

Reviewer #1: Yes

Reviewer #2: Yes

4. Is the manuscript presented in an intelligible fashion and written in standard English?

Reviewer #1: Yes

Reviewer #2: Yes

Reviewer #1: The manuscript is technically sound, the author followed the guidelines in submitting their manuscript. The methods used to select the participants is clearly mentioned although some were purposively selected which introduces selection bias but the author has mentioned this as a limitation in the study.

The information reads well but the level of education or cadres for the community pharmacy health care practitioners is not clear. This would further inform the reader about the competences and knowledge the community pharmacy health care practitioners have about TB.

Reviewer #2: The manuscript presents a theory-informed qualitative study designed to explore the feasibility of integrating tuberculosis treatment into community pharmacies for TB/HIV patients in Kampala, Uganda. Through interviews with patients, healthcare providers from six public health facilities, and Ministry of Health experts, the study identified facilitators and barriers to integration. The findings suggest that integrating TB treatment into community pharmacies offers advantages such as convenience, privacy, and cost-effectiveness, but also faces challenges like inadequate pharmacy infrastructure, insufficient staff training, and unclear operational procedures. I have some comments for future improvement.

1. In the “Methods” section, the authors mention using the saturation principle to determine the sample size, but the description of how to specifically operate this principle is not detailed. For example, the text mentions “Specifically, this was observed when ≤5% of the information was new, conditional on a base size of 20 and a run length of 15,” but it does not explain how to judge the novelty of the information and how to decide whether saturation has been reached based on this standard in actual operation.

2. In the “Methods” section, the authors mention using framework analysis for data analysis, but the description of the coding process is not clear. For example, the text mentions “JI and RK independently reviewed 5-7 transcripts to familiarize themselves with the data, coded the transcripts by flagging important texts, identified emerging patterns, and jointly developed a preliminary codebook by consensus,” but it does not provide a detailed explanation of how to determine which texts are “important” and how to identify “emerging patterns.”.

3. In the “Discussion” section, the authors discuss the research results, but the interpretation of some key results is not deep enough. For example, the text mentions “Enhanced anonymity provided by community pharmacies emerged as a key facilitator of TB treatment integration,” but it does not further explore how this anonymity specifically affects patients' behavior and psychology and the potential mechanisms of its impact on treatment outcomes.

4. In the “Conclusion and recommendation” section, the authors propose research recommendations, but the recommendations are not specific enough. For example, the text mentions “context-specific interventions that are developed in collaboration with key stakeholders, address barriers such as eligibility criteria refinement, enhanced pharmacy HCP training, and financial incentives, and leverage facilitators like policy support and stakeholder readiness, are needed,” but it does not provide detailed suggestions on how to implement these interventions. It is suggested that the authors supplement more specific implementation suggestions.

**Do you want your identity to be public for this peer review?** For information about this choice, including consent withdrawal, please see our Privacy Policy

Reviewer #1: No

Reviewer #2: **Yes: ** Pengpeng Ye

---

## [Decision Letter · Decision Letter 1]

10 Nov 2025

Barriers and facilitators to integrating tuberculosis treatment into community pharmacies for people with TB/HIV in Kampala, Uganda: a qualitative study

PGPH-D-25-01867R1

Dear Dr. Izudi,

We are pleased to inform you that your manuscript 'Barriers and facilitators to integrating tuberculosis treatment into community pharmacies for people with TB/HIV in Kampala, Uganda: a qualitative study' has been provisionally accepted for publication in PLOS Global Public Health.

Best regards,

Henry Zakumumpa, PhD

Academic Editor

Reviewer Comments (if any, and for reference):

Reviewer's Responses to Questions

**Comments to the Author**

Reviewer #1: All comments have been addressed

Reviewer #2: All comments have been addressed

publication criteria?

Reviewer #1: Yes

Reviewer #2: Yes

3. Has the statistical analysis been performed appropriately and rigorously?

Reviewer #1: Yes

Reviewer #2: N/A

4. Have the authors made all data underlying the findings in their manuscript fully available (please refer to the Data Availability Statement at the start of the manuscript PDF file)?

Reviewer #1: Yes

Reviewer #2: Yes

5. Is the manuscript presented in an intelligible fashion and written in standard English?

Reviewer #1: Yes

Reviewer #2: Yes

Reviewer #1: All comments have been addressed.

Reviewer #2: I have no additional comments.

**Do you want your identity to be public for this peer review?** For information about this choice, including consent withdrawal, please see our Privacy Policy

Reviewer #1: No

Reviewer #2: **Yes: ** Pengpeng Ye
